# 3D Model-Guided Robot-Assisted Giant Presacral Ganglioneuroma Exeresis by a Uro-Neurosurgeons Team: A Case Report

**DOI:** 10.3390/reports8030099

**Published:** 2025-06-20

**Authors:** Leonardo Bradaschia, Federico Lavagno, Paolo Gontero, Diego Garbossa, Francesca Vincitorio

**Affiliations:** 1Neurosurgery Unit, Department of Neuroscience “Rita Levi Montalcini”, A.O.U. Città della Salute e della Scienza, University of Turin, 10127 Turin, Italy; diego.garbossa@unito.it (D.G.); vincitorio.francesca@gmail.com (F.V.); 2Urology Unit, Department of General & Specialized Surgery, A.O.U. Città della Salute e della Scienza, University of Turin, 10127 Turin, Italy; federico.lavagno@unito.it (F.L.); paolo.gontero@unito.it (P.G.)

**Keywords:** laparotomy, laparoscopy, retroperitoneal, case report, nervous monitoring

## Abstract

**Background and Clinical Significance:** Robotic surgery reduces the need for extensive surgical approaches and lowers perioperative complications. In particular, it offers enhanced dexterity, three-dimensional visualization, and improved precision in confined anatomical spaces. Pelvic masses pose significant challenges due to their close relationship with critical neurovascular structures, making traditional open or laparoscopic approaches more invasive and potentially riskier. Robot-assisted resection, combined with intraoperative neurophysiological monitoring, may therefore offer a safe and effective solution for the management of complex pelvic lesions. **Case Presentation:** An 18-year-old woman was incidentally diagnosed with an 11 cm asymptomatic pelvic mass located anterior to the sacrum. Initial differential diagnoses included neurofibroma, teratoma, and myelolipoma. Histopathological examination confirmed a ganglioneuroma. Following multidisciplinary discussion, the patient underwent a robot-assisted en bloc resection using the Da Vinci Xi multiport system. Preoperative planning was aided by 3D modeling and intraoperative navigation. **Conclusions:** Surgery lasted 322 min. Preoperative and postoperative eGFR values were 145.2 mL/min and 144.0 mL/min, respectively. The lesion measured 11 cm × 9 cm × 8 cm. The main intraoperative complication was a controlled breach of the iliac vein due to its close adherence to the mass. No major postoperative complications occurred (Clavien-Dindo Grade I). The drain was removed on postoperative day 3, and the bladder catheter on day 2. The patient was discharged on postoperative day 5 without further complications. Presacral ganglioneuromas are rare neoplasms in a surgically complex area. A multidisciplinary approach using robotic-assisted laparoscopy with nerve monitoring enables safe, minimally invasive resection. This strategy may help avoid open surgery and reduce the risk of neurological and vascular injury.

## 1. Introduction and Clinical Significance

Ganglioneuromas are benign, well-differentiated neoplasms of the sympathoadrenal nervous system, and can therefore arise anywhere along the sympathetic chain. They are slightly more common in young female patients [1,2]. A pelvic localization is considered rare, and when these tumors occupy the presacral retroperitoneal space—adjacent to the ureters and iliac vessels—surgical access often requires wide exposure and significant retraction of the peritoneal organs.

In recent years, robotic surgery has gained popularity due to its ability to minimize the need for extensive surgical approaches. It offers reduced postoperative pain, morbidity, and recovery time, potentially making complex dissections more manageable compared to conventional laparoscopy.

One of the main challenges in pelvic surgery is the proximity to the lumbosacral plexus, which may be displaced or injured during dissection. Intraoperative neurophysiological monitoring can help minimize the risk of nerve damage and improve surgical outcomes.

We present the case of a presacral ganglioneuroma treated with robot-assisted surgery using the da Vinci^®^ (Intuitive Surgical, Sunnyvale, CA, USA) platform by a multidisciplinary team of neurosurgeons and urologists, with the intraoperative support of a specialized neurophysiologist for lumbosacral plexus monitoring. Preoperative planning included 3D reconstruction of the lesion using Brainlab^®^ (BrainLAB AG, Munich, Germany) software, which allowed for enhanced visualization of the tumor’s anatomical relationships with surrounding structures.

## 2. Case Presentation

An 18-year-old woman presented to our clinic in March 2024 after undergoing pelvic contrast-enhanced Magnetic Resonance Imaging (MRI) the previous month on the recommendation of her gynecologist. The patient had been experiencing dysmenorrhea for the past two years and sought consultation to start contraceptive therapy. During an ultrasound examination, an incidental mass was detected, initially suspected to be an ovarian cyst. However, the MRI revealed a sacrococcygeal mass measuring approximately 106 mm × 77.4 mm × 49.8 mm, extending from S1 to the coccyx (Figure 1A–C). The differential diagnosis included neurofibroma, teratoma, and myelolipoma.

At clinical evaluation, the patient was completely asymptomatic, with no urinary or bowel impairments. Dysmenorrhea resolved after starting the contraceptive pill. The patient had no other comorbidities nor past medical/surgical history, and there was no family history of neurofibromatosis. The contraceptive pill was her only medical therapy. Following a multidisciplinary discussion between urologists and neurosurgeons, a robot-assisted en bloc excision of the lesion was planned. In this specific case, the robotic approach was preferred not only for its general advantages, but also because it allowed for precise dissection in a deep and anatomically complex pelvic region, where the lesion was in close proximity to critical neurovascular structures. The patient was an 18-year-old woman, in full reproductive age, which made the preservation of reproductive organs and function a priority. Moreover, the minimally invasive nature of robotic surgery ensured better cosmetic outcomes—an important consideration in young patients—while reducing the risk of surgical trauma compared to open procedures. The enhanced visualization and instrument dexterity provided by the robotic platform were therefore essential in achieving a safe and function-sparing resection.

An Angio-CT scan was performed preoperatively (Figure 2) and used for 3D reconstruction of the vascular structures and their relationship with the lesion, utilizing Brainlab^®^ (BrainLAB AG, Germany) planning software (Figure 3). Additionally, a 3D digital reconstruction of the pelvic mass, with a total volume of 240 cm^3^, was created from the contrast-enhanced MRI and fused with the Angio-CT data, allowing for precise 3D planning of the tumor’s position relative to surrounding vascular structures.

In July 2024, the patient was hospitalized for surgery. Intraoperative Neurophysiological Monitoring (IONM) was performed using standard techniques with remote channel amplifiers and stimulators operated by dedicated neurophysiologists and IONM technicians. Subdermal corkscrew electrodes were positioned over the somatosensory and motor cortices, while needle electrodes were placed on the gastrocnemius, tibialis anterior, quadriceps, abductor hallucis, and external anal sphincter muscles; pudendal nerve activity and the bulbocavernosus reflex were also monitored. Anesthesia was optimized for recording Somatosensory Evoked Potentials (SSEPs), Motor Evoked Potentials (MEPs), and Electroencephalography (EEG). A nerve stimulator was used for direct stimulation during the dissection of the lesion’s borders (Figure 4D).

After establishing pneumoperitoneum, the robotic system (da Vinci^®^ Xi, Intuitive Surgical, Sunnyvale, CA, USA) was docked using a standard pelvic configuration. The uterus and ovaries were suspended to improve exposure, and the posterior peritoneum was opened (Figure 4A). The right ureter and common iliac vessels were carefully dissected to expose the cranial portion of the mass (Figure 4B,C), which appeared well-encapsulated and firm. During dissection of the lateral pseudocapsule, a small tear of the right internal iliac vein occurred and was successfully repaired with a 4/0 monofilament suture (Figure 4E).

The dissection continued in a cranio-caudal direction using the pseudocapsule as a cleavage plane, and the left-posterior margin was freed. The mass extended to the coccyx posteriorly, and a dissection plane between the rectum and the tumor was identified, enabling safe resection without damaging the rectum. However, IONM indicated possible involvement of the sacral nerve roots, detected through direct stimulation with an external anal sphincter (EAS) response. Consequently, resection was halted at the most caudal part of the tumor to avoid neurological deficits (Figure 4F). Figure 4(**A**) After establishing pneumoperitoneum, the peritoneum was incised at the junction of the right ureter and the iliac vessels. (**B**) Exposure of the right ureter. (**C**) The pseudocapsule was carefully dissected from the surrounding tissue. (**D**) Intraoperative neuromonitoring was instrumental in avoiding injury to the sacral plexus. Direct nerve stimulation was combined with MEPs, SSEPs, and EEG monitoring. (**E**) During dissection, a small tear in the right internal iliac vein occurred and was repaired using a 4/0 monofilament suture. (**F**) Resection was halted at the most caudal part of the tumor to prevent neurological deficits, following IONM detection of nearby neural structures.
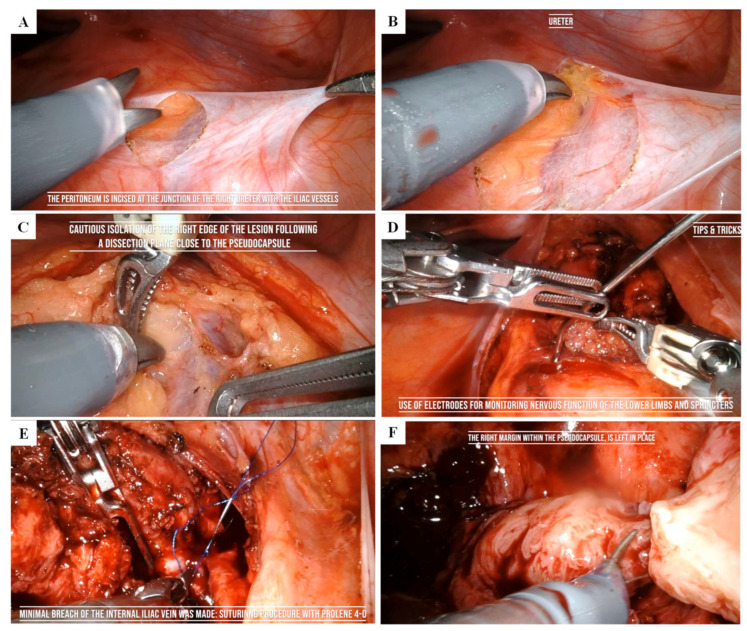


Once the lesion was completely mobilized, it was placed in an endobag. Hemostasis was achieved using rayon gauze (Tabotamp^®^), and the specimen was extracted through a Pfannenstiel incision. The total operative time was 322 min, with approximately 200 min spent at the console. The estimated blood loss was minimal, around 500 mL.

Macroscopic examination revealed a firm, encapsulated mass measuring 11 cm × 9 cm × 8 cm (Figure 5A), with a smooth surface. On sectioning, the tumor appeared gray-yellowish, with a fibrous capsule on the cranial and anterior parts (Figure 5B).

**Figure 5 reports-08-00099-f005:**
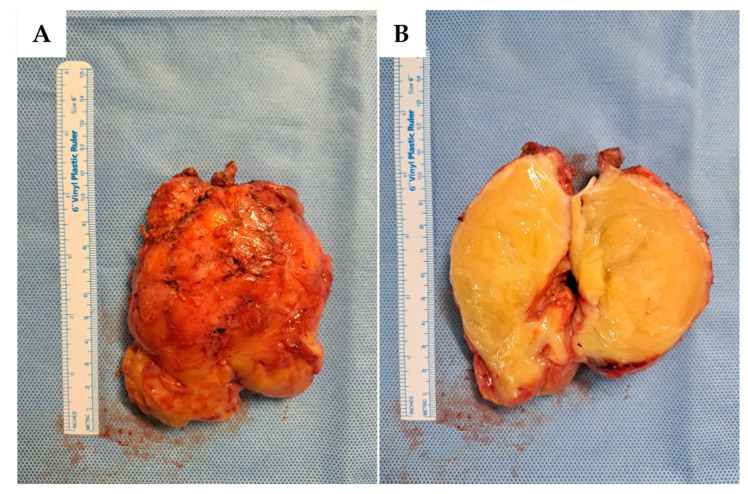
(**A**) Macroscopic examination of the excised specimen. It appeared as a firm, encapsulated mass measuring 11 cm × 9 cm × 6.5 cm, with a smooth surface. (**B**) Macroscopic examination of the excised specimen. On sectioning, the mass appeared gray-yellowish in color, with a recognizable fibrous capsule in the cranial and anterior portions.

Microscopic analysis showed Schwannian stromal tissue interspersed with mature ganglion cells and some immature cells (Figure 6A–C). Immunohistochemistry revealed diffuse S-100 positivity in the Schwannian stroma, CD34 negativity, focal PHOX2B positivity in the immature ganglion cells, and CD3 positivity in small T-cell aggregates (Figure 6D). The final diagnosis was ganglioneuroma, maturating subtype, according to the International Neuroblastoma Pathology Classification (INPC).

Preoperative eGFR was 145.2 mL/min, Hb: 11.2 g/dL and Hct: 32.9% and postoperative eGFR was 144.0 mL/min, Hb: 10.3 g/dL and Hct 29.8%. On postoperative day 2, the urinary catheter was removed without signs of urinary retention, and the surgical drain was removed on day 3. The patient was discharged on day 4 without neurological deficits or pain (Clavien-Dindo grade I).

A follow-up contrast-enhanced MRI performed 30 days after surgery revealed a small residual lesion at the caudal portion of the surgical site, near the coccyx, consistent with remaining biological tumor tissue. In addition, a fluid-filled area was observed, likely related to the patient’s ovulatory phase and not indicative of pathological findings (Figure 7). No other pathological changes were observed. On the 30-day outpatient follow-up, the patient had resumed normal daily activities without pain or neurological deficits. A 3-month follow-up contrast-enhanced pelvic MRI was scheduled due to the benign nature of the tumor.

Written informed consent was obtained from the patient for the publication of the present case report and any accompanying images.

## 3. Discussion

### 3.1. Ganglioneuroma

Ganglioneuromas belong to the family of Peripheral Neuroblastic Tumors (pNTs), which also includes neuroblastomas and ganglioneuroblastomas [3]. Unlike neuroblastomas, ganglioneuromas are considered to be tumors at the final stage of maturation within the pNT spectrum and, therefore, benign [4]. Their origin from the neural crest explains their composition of mature (or maturing) ganglion cells intermixed with abundant nerve fibers, predominantly unmyelinated, and associated Schwann cells [5]. These tumors typically occur in female children and young adults [6], with a predilection for sites where sympathetic ganglia are located, including the posterior mediastinum (41.5%), retroperitoneum (37.5%), adrenal medulla (21%), and the paraspinal region, particularly in the neck (8%) [7,8]. A presacral location is extremely rare, with fewer than 20 reported cases in the literature as of 2013 [9].

In 1986, an international consensus group established the International Neuroblastoma Staging System (INSS), based on clinical, radiological, and surgical criteria [10]. According to this system, the lesion reported in this article would be classified as stage I. Surgical excision is typically the treatment of choice, with no additional need for radiotherapy or chemotherapy.

### 3.2. Pelvic Region

The pelvic region is an anatomically complex space that contains numerous structures within a small volume, particularly in females. Moreover, multiple specialists—including gynecologists, urologists, general surgeons, vascular surgeons, neurosurgeons, and orthopedic surgeons—are involved in managing this area, each focusing on their respective pathologies. As such, the management of a newly discovered mass in the pelvic region may differ depending on its anatomical relationship with surrounding organs. In some cases, a multidisciplinary approach is required, especially for neoplasms that may fall under neurosurgical competence but are located in areas traditionally managed by other specialists.

### 3.3. Considerations

While most Ganglioneuromas (GNs) are diagnosed in children and adolescents, adult cases—especially those involving the presacral or pelvic space—are exceptional. These tumors often reach considerable size before becoming symptomatic, owing to their slow growth and deep anatomical location.

Several reports in the literature have documented giant ganglioneuromas in adults, often exceeding 10 cm in size and displacing or encasing critical retroperitoneal structures. Kirchweger et al. [11] presented a case in 2020 involving both retroperitoneal and mediastinal extension, emphasizing the technical challenge of complete resection and the importance of multidisciplinary management. Feng and Wang [12] described in 2023 a large retroperitoneal GN evaluated preoperatively with advanced multimodal ultrasound imaging, demonstrating the value of precise imaging in surgical planning. Similarly, Lebby et al. [13] reported in 2021 a retroperitoneal GN in an HIV-positive adult with nodal involvement, underlining diagnostic uncertainty in immunocompromised patients. More recently, Bapir et al. [14] discussed a diagnostically challenging case of a large retroperitoneal ganglioneuroma in a middle-aged patient, highlighting surgical considerations in anatomically deep and confined spaces.

These cases underscore the technical complexity and variability of adult-onset ganglioneuromas, particularly when large in size or located in anatomically constrained regions such as the pelvis or presacral space. In such scenarios, the use of minimally invasive techniques, including robot-assisted surgery, may offer enhanced visualization, greater precision, and a lower risk of damage to surrounding neurovascular and reproductive structures—especially in younger patients, where functional preservation is crucial.

In the case described, the mass was located in the retroperitoneal space, anterior to the sacrum and extending from S1 to the coccyx, in close contact with the sigmoid colon, rectum, right ureter, and right internal iliac vein. Given its location and the absence of symptoms, several management strategies were considered, including conservative treatment with close follow-up [15]. However, due to the lack of a definitive histopathological diagnosis and the initial consideration of the lesion as a neurofibroma, the potential risk of malignancy led to the exclusion of the conservative approach; laparotomy excision, would allow for a more rapid surgical procedure with direct visualization of the involved structures, either through a transperitoneal approach [9,16,17,18,19,20] or an extraperitoneal approach [21,22]. However, given the patient’s young age and the possibility of performing minimally invasive surgery, laparoscopic surgery was deemed more appropriate. Initially, standard laparoscopy was considered [23,24,25], but after discussions with the urology team at our institution, a robot-assisted laparoscopy was selected instead.

To the best of our knowledge, only two other cases have been reported in the literature involving a robot-assisted approach for the resection of presacral ganglioneuroma: Palep et al. in 2015 [26] and Garzon-Muvdi et al. in 2019 [27]. In Palep et al.’s study, the importance of a multidisciplinary team, including consultation with a spine neurosurgeon, was emphasized, while in Garzon-Muvdi et al.’s report, intraoperative nerve monitoring was implemented. In our case, we combined both approaches, using a co-joint neurosurgical–urologic team along with complete nerve monitoring, provided by a specialized neurophysiologist present during the surgery. Our operating time was approximately the same as that reported by Palep et al. (console time: 200 vs. 195 min), while it was significantly shorter than the time reported by Garzon-Muvdi et al. (total surgical time: 5 h vs. 9 h). In all cases, blood loss was minimal (40 mL in Palep et al., 200 mL in Garzon-Muvdi et al., 500 mL in our case), and all patients were discharged within a few days after surgery (3 days in Palep et al., 7 days in Garzon-Muvdi et al., 3 days in our case), with minimal postoperative symptoms. Notably, Garzon-Muvdi et al. reported a left-sided dorsiflexion weakness that spontaneously recovered during a 6-month follow-up, as well as a right-sided neuropathic pain in the S2 and S3 distribution, which had been present prior to surgery.

Exiting the presacral region, Tavares et al. [28] reported a combined approach involving robotic excision via an anterior/retroperitoneal route and a posterior hemilaminectomy/microsurgical technique to achieve complete resection of a paravertebral ganglioneuroma centered on the right D12–L1 foramina and extending into the spinal canal. This case highlights the potential of robot-assisted procedures as adjunct therapies for lesions in anatomically complex locations.

Regarding neuroblastic tumors as a whole—i.e., including neuroblastomas, ganglioneuroblastomas, and ganglioneuromas—their occurrence in adults is extremely rare and often presents with nonspecific symptoms, delayed diagnosis, and therapeutic uncertainty, particularly when located in unusual anatomical sites.

Telecan et al. [29] described in 2022 a rare case of primary adrenal neuroblastoma in a 34-year-old adult, illustrating the diagnostic and therapeutic complexities of neuroblastic tumors in this age group. Despite originating from the adrenal gland—a more common site in children—the tumor in this adult patient showed aggressive features and required extensive surgical and oncological management. The authors emphasized the absence of standardized treatment guidelines for adults and the need to individualize care based on tumor biology and extent.

Other reports, such as Godkhindi et al. [30], have described large retroperitoneal neuroblastomas in young adults, reinforcing the notion that these tumors can arise in atypical locations and may mimic more common neoplasms or benign masses, thereby complicating diagnosis and management. In rare cases, even malignant transformation of benign neuroblastic lesions has been documented, such as in the report by Kulkarni et al. [31], which described a spinal neuroblastoma arising from a previously diagnosed ganglioneuroma.

These cases collectively highlight the diagnostic ambiguity, surgical complexity, and biological unpredictability of neuroblastic tumors in the adult population. They support the rationale for choosing surgical approaches that maximize anatomical precision and minimize collateral damage, such as robot-assisted resection with intraoperative neuromonitoring, especially when tumors are located near critical neurovascular and reproductive structures.

## 4. Conclusions

Pelvic ganglioneuromas are rare benign tumors that may arise in anatomically complex regions such as the presacral space. In selected cases, robot-assisted surgery may represent a safe and effective approach for managing these retroperitoneal neoplasms. This technique can potentially offer shorter operative times compared to conventional laparoscopy and similar durations to standard median laparotomy, with minimal blood loss and low postoperative morbidity. Moreover, the ability to achieve en bloc resection may be particularly valuable in cases involving more aggressive lesions, such as malignant peripheral nerve sheath tumors (MPNSTs) or aggressive neurofibromas. Finally, the integration of intraoperative neuromonitoring (IONM) of the lumbosacral plexus throughout the procedure may contribute to safer tumor dissection and a reduced risk of postoperative neurological complications.

## Figures and Tables

**Figure 1 reports-08-00099-f001:**
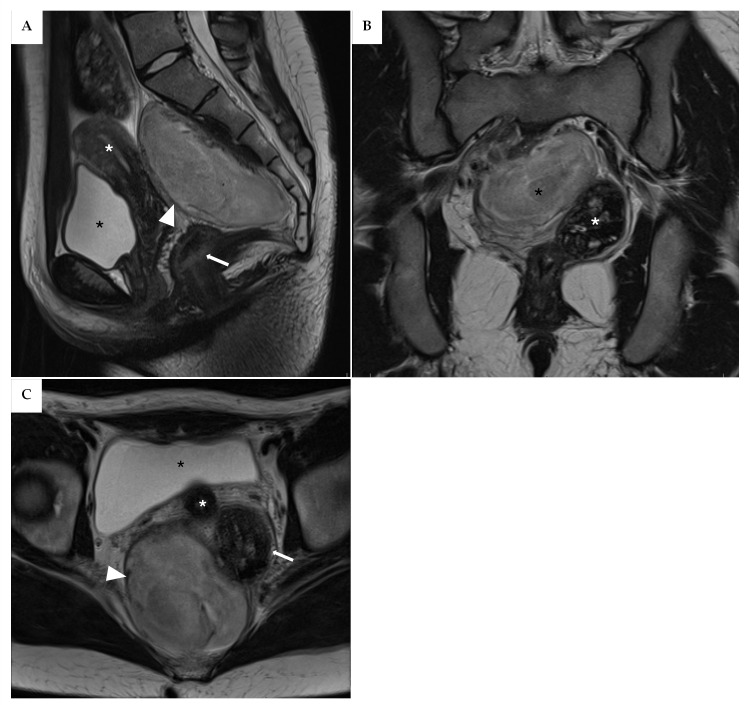
(**A**) Mid-sagittal TSE T2-weighted image. A voluminous mass is visible in the presacral space, extending from S1 to the coccyx (white arrowhead). There is initial compression of the rectum without associated clinical symptoms (white arrow), as well as mild compression of the uterus (white asterisk). The bladder shows no involvement (black asterisk). (**B**) Coronal TSE T2-weighted image. The presacral mass (black asterisk) appears to have a well-defined border on the left side, while on the right side it is not clearly distinguishable. White asterisk: rectum. (**C**) Axial TSE T2-weighted image. The presacral mass (white arrowhead) is in contact with the rectum (white arrow), with no clear cleavage plane visible on imaging; however, a plane of separation was fortunately identified during surgery. White asterisk: uterus. Black asterisk: bladder.

**Figure 2 reports-08-00099-f002:**
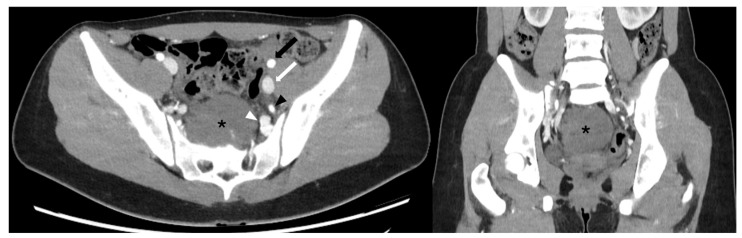
Angio-CT scan, axial (**left**) and coronal (**right**) views. The presacral mass (black asterisk) is in close contact with vascular structures, including the internal iliac vein (white arrowhead) and internal iliac artery (black arrowhead). In contrast, the external iliac artery (black arrow) and vein (white arrow) are spared.

**Figure 3 reports-08-00099-f003:**
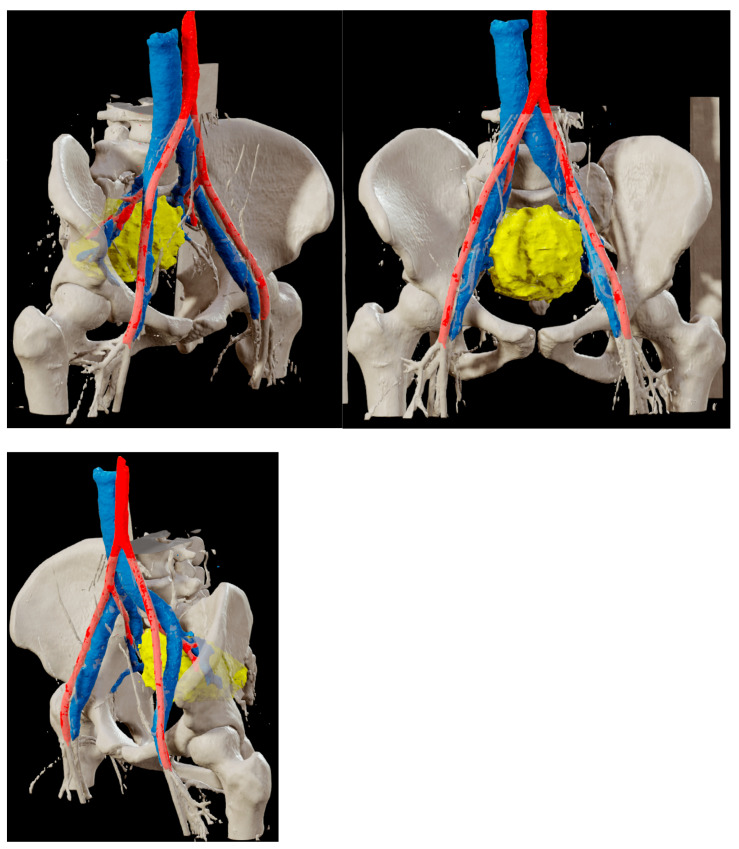
Digital 3D reconstruction of the presacral mass (yellow) and its relationship with the internal iliac vein (blue) and artery (red), based on Angio-CT scan data fused with contrast-enhanced MRI using Brainlab^®^ planning software (BrainLAB AG, Germany). The reconstruction was utilized both preoperatively and intraoperatively to enhance the three-dimensional understanding of the lesion.

**Figure 6 reports-08-00099-f006:**
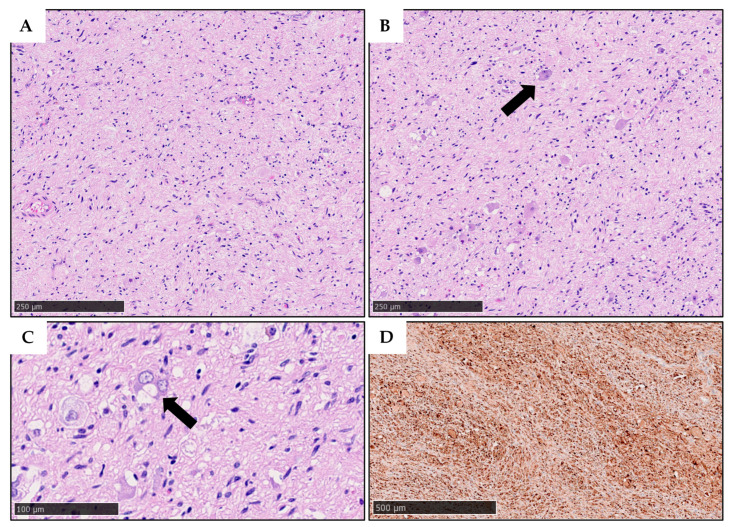
Histopathological findings of the resected lesion. (**A**,**B**) Low-power images (HE, original magnification: 100×) showing a proliferation comprising prevalent Schwannian stroma (**A**) with mixed ganglion cells (**B**), arrow: a binucleated ganglion cell. (**C**) Scattered ganglion elements showed incomplete maturation with a higher nuclear/cytoplasmic ratio (HE, original magnification: 200×). (**D**) Widespread expression of S100 was observed as expected due to the prevalence of the Schwannian stroma (S100 immunohistochemistry, original magnification: 40×). These findings were deemed consistent with a diagnosis of ganglioneuroma, maturing subtype (according to the INPC classification).

**Figure 7 reports-08-00099-f007:**
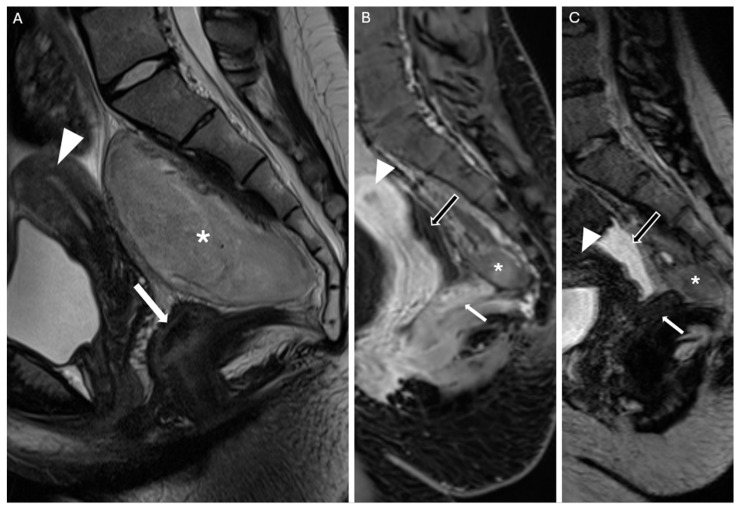
(**A**) Mid-sagittal TSE T2-weighted image from the preoperative MRI for comparison. (**B**) Postoperative contrast-enhanced MRI at 30 days: mid-sagittal T1 VIBE Dixon-enhanced image. A small residual portion of the neoplasm is visible anterior to the coccyx (white asterisk); the patient reported no symptoms during the outpatient follow-up. At the time of the MRI, the patient was in the ovulatory phase, which explains the presence of fluid in the Douglas pouch (black arrow with white outline). (**C**) Postoperative contrast-enhanced MRI at 30 days: same mid-sagittal view as in image B, T2 SPACE-weighted image. White arrowhead: uterus; white arrow: rectum.

## Data Availability

Due to patient privacy concerns, the data presented in this study are available from the corresponding author upon reasonable request.

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
