# Peer review of "3D Model-Guided Robot-Assisted Giant Presacral Ganglioneuroma Exeresis by a Uro-Neurosurgeons Team: A Case Report"

_reports, 2025, doi:10.3390/reports8030099_

Round 1

Reviewer 1 Report

Comments and Suggestions for Authors

The article presents the case clearly and concisely. From the introduction and discussion, it caught my attention that authors describe ganglioneuroma as more common in female than in males, when its frequency varies among the studies and the relationship is not clear. 

In the discussion, authors say that the risk of potential malignancy ruled out conservative treatment. However, the citation (9) presents a case that showed no signs of malignancy during 6 years of follow up. Ganglioneuroma is the most mature tumor / tissue within the neuroblastoma - ganglioneuroma spectrum and, as a general rule, they do not become malignant. Therefore, observation should be considered a valid management approach. 

Author Response

Comments 1: “From the introduction and discussion, it caught my attention that authors describe ganglioneuroma as more common in female than in males, when its frequency varies among the studies and the relationship is not clear.”

Response 1: “Thank you for this valuable comment. We agree that the literature presents some variability regarding the sex distribution of ganglioneuromas. However, in response to your observation, we have revised the text to better reflect this nuance and have cited two that report a slight female predominance (citation 1 and 2). These references have been added to support the revised statement and to acknowledge the variability across studies.

We hope this clarification addresses your concern.”

Comment 2: “In the discussion, authors say that the risk of potential malignancy ruled out conservative treatment. However, the citation (9) presents a case that showed no signs of malignancy during 6 years of follow up. Ganglioneuroma is the most mature tumor / tissue within the neuroblastoma - ganglioneuroma spectrum and, as a general rule, they do not become malignant. Therefore, observation should be considered a valid management approach. “

Response 2: "Thank you for your insightful comment. We fully agree that ganglioneuromas are considered benign and are the most differentiated tumors within the neuroblastic spectrum, with malignant transformation being extremely rare.

In our case, however, the initial radiological suspicion favoured a neurofibroma rather than a ganglioneuroma, and thus a conservative approach was not considered appropriate due to the potential—albeit low—risk of malignant progression associated with peripheral nerve sheath tumors.

Furthermore, given the patient’s young age and the potential desire for future pregnancy, surgical removal was favoured to avoid future diagnostic uncertainties or complications during gestation. We have clarified this point in the revised manuscript to better reflect the rationale behind our management decision.”

Reviewer 2 Report

Comments and Suggestions for Authors

This case report describes the robot-assisted resection of a large presacral ganglioneuroma in an 18-year-old woman, performed by a multidisciplinary team of neurosurgeons and urologists. The authors detail the use of advanced preoperative 3D modeling, intraoperative neurophysiological monitoring, and the da Vinci Xi robotic platform to achieve a safe, minimally invasive excision of the tumor. The report is well-illustrated, methodically structured, and offers valuable insight into the technical considerations and benefits of a robotic approach in managing complex pelvic tumors. The case adds to the limited literature on robot-assisted surgery for presacral ganglioneuromas and is especially notable for its comprehensive use of modern surgical planning and nerve preservation techniques. Please find bellow a point-by-point suggestion list of further refining the manuscript:

Title and Abstract

  • Consider including “presacral” in the title to emphasize the anatomical complexity.

  • In the abstract, clarify why the robotic approach was favored over traditional surgery.

Introduction

  • Expand the literature review to include other adult-onset neuroblastic tumors.

  • Suggest citing:
    Telecan T, et al. Adrenal Gland Primary Neuroblastoma in an Adult Patient: A Case Report and Literature Review. Medicina. 2022;59(1):33.

Case Presentation / Materials & Methods

  • Add a clearer narrative timeline from diagnosis to treatment.

  • Briefly explain the rationale for preferring robotic surgery beyond its general advantages.

Surgical Technique

  • Consider shortening procedural details (trocar placement, docking) to reduce redundancy.

  • Add intraoperative pictures of the representative surgical steps.

Results

  • Clarify whether the residual lesion on follow-up imaging is biological tumor tissue or postoperative fluid.

Discussion

  • Broaden the literature review to include similar neuroblastic tumors in other adult locations.

  • Incorporate the 2022 case report to support the rarity and complexity of adult neuroblastic tumors.

Conclusion

  • Adjust tone slightly to avoid overgeneralizing based on a single case.

Author Response

Comment 1: “Consider including “presacral” in the title to emphasize the anatomical complexity.”

Response 1: “Thank you for the helpful suggestion. We have revised the title to include the term “presacral” in order to highlight the anatomical complexity of the case, as recommended.”

Comment 2: “In the abstract, clarify why the robotic approach was favored over traditional surgery.”

Response 2: “Thank you for your comment. To maintain the abstract within the recommended length and avoid excessive detail, we did not include the rationale for choosing the robotic approach there. However, we have added a clear explanation in the main text, outlining the advantages of the robotic technique in terms of precision, visualization, and access in anatomically complex regions such as the presacral space (…the minimally invasive nature of robotic surgery ensured better cosmetic outcomes—an important consideration in young patients—while reducing the risk of surgical trauma compared to open procedures. The enhanced visualization and instrument dexterity provided by the robotic platform were therefore essential in achieving a safe and function-sparing resection).”

Comment 3: “Expand the literature review to include other adult-onset neuroblastic tumors. Suggest citing: Telecan T, et al. Adrenal Gland Primary Neuroblastoma in an Adult Patient: A Case Report and Literature Review. Medicina. 2022;59(1):33.”

Response 3: “Thank you for your thoughtful suggestion. We have expanded the literature review to include the recommended article by Telecan et al., along with additional references addressing adult-onset neuroblastic tumors.

While we acknowledge the importance of contextualizing ganglioneuroma within the broader neuroblastic tumor spectrum, we have intentionally focused this manuscript on ganglioneuromas, given their distinct clinical behaviour and relevance to the presented case. We have aimed to maintain this focus while still incorporating relevant references to support a broader understanding where appropriate. We hope this approach meets your expectations.”

Comment 4: “Add a clearer narrative timeline from diagnosis to treatment.”

Response 4: “Thank you for the suggestion. We have reviewed the manuscript and confirmed that a narrative timeline is already provided, stating that the patient was referred to our unit in March 2024, underwent surgical treatment in June 2024, and was discharged on postoperative day 4.

We believe this sequence offers a clear and concise overview of the diagnostic and therapeutic process. To preserve patient anonymity, we have chosen not to include additional time-related details that could potentially make the case identifiable.”

Comment 5: “Briefly explain the rationale for preferring robotic surgery beyond its general advantages.”

Response 5: “Thank you for this comment. As suggested, we have addressed the specific rationale for choosing the robotic approach in the manuscript. In particular, we added “In this specific case, the robotic approach was preferred not only for its general advantages, but also because it allowed for precise dissection in a deep and anatomically complex pelvic region, where the lesion was in close proximity to critical neurovascular structures”.“

Comment 6: “Consider shortening procedural details (trocar placement, docking) to reduce redundancy.”

Response 6: “We have revised the surgical section accordingly by shortening the descriptions of trocar placement and docking to avoid redundancy, while retaining the essential procedural information.”

Comment 7: “Add intraoperative pictures of the representative surgical steps.”

Response 7: “Thank you for the suggestion. As requested, we have added Figure 5, which illustrates the key intraoperative steps of the lesion excision. We believe these images help clarify the surgical approach and enhance the educational value of the report.”

Comment 8: “Clarify whether the residual lesion on follow-up imaging is biological tumor tissue or postoperative fluid.”

Response 8: “Thank you for this important observation. In the revised manuscript, we have clarified that the follow-up imaging revealed a small residual component, part of which is consistent with residual tumor tissue, while another portion appears to be fluid, most likely related to the patient's ovulatory phase.

Comment 9: “Broaden the literature review to include similar neuroblastic tumors in other adult locations.”

Response 9: “Thank you for the constructive suggestion. We have expanded the literature review to include additional reports of neuroblastic tumors in adult patients, with a particular focus on ganglioneuromas located in various anatomical regions. This broader perspective helps contextualize the rarity and behaviour of these lesions while preserving the main focus of our manuscript on presacral ganglioneuroma.”

Comment 10: “Adjust tone slightly to avoid overgeneralizing based on a single case.”

Response 10: “We have reviewed the manuscript carefully and adjusted the tone in several sections to avoid overgeneralization based on a single case. The revised text now better reflects the nature and limitations of case-based evidence.

We appreciate your guidance in strengthening the manuscript's scientific rigor.”

Reviewer 3 Report

Comments and Suggestions for Authors

Congratulations for your interesting case report. The manuscript is well-written and presented. Some suggestions for imrpovement:

  • Please include the manufacturer on the first mentioning of DaVinci (line 48 instead of line 108)
  • I would include some more laboratory tests to prove the minor blood loss (Hct/Hb before and after surgery for example)
  • I would present the part medical and surgical history as well as the medications of the patient at the beginning of the case report description.
  • I would avoid the separation of the Discussion under subtitles. However, the structure is complete and well-presented, making the subtitle admissible.

Author Response

Comment 1: “Please include the manufacturer on the first mentioning of DaVinci (line 48 instead of line 108)”

Response 1: “Thank you for the suggestion. We have updated the manuscript to include the manufacturer at the first mention of the Da Vinci system (line 48), as requested.”

Comment 2: “I would include some more laboratory tests to prove the minor blood loss (Hct/Hb before and after surgery for example)”

Response 2: “Thank you for this helpful suggestion. We have added pre- and postoperative haemoglobin and haematocrit values, as well as the total estimated blood loss during surgery, to better document the minimal blood loss.”

Comment 3: “I would present the part medical and surgical history as well as the medications of the patient at the beginning of the case report description.”

Response 3: “Thank you for the suggestion. We have added a clear statement at the beginning of the case report indicating that the patient had no comorbidities, no past medical or surgical history, and no family history of neurofibromatosis. This addition aims to provide a more comprehensive clinical background as requested.”

Comment 4: “I would avoid the separation of the Discussion under subtitles. However, the structure is complete and well-presented, making the subtitle admissible.”

Response 4: “Thank you for your feedback. We appreciate your point regarding the use of subtitles in the Discussion section. We have chosen to maintain the subdivision as it enhances clarity and allows for a more organized and focused discussion of the various aspects of the case. We hope this structure improves readability without compromising the flow of the manuscript.”

Round 2

Reviewer 2 Report

Comments and Suggestions for Authors

Thank you for taking the suggestions into consideration. I congratulate the authors for the current version of the manuscript and I find it suitable for publication.